# Influence of the COVID-19 Pandemic on Dental Emergency Admissions in an Urgent Dental Care Service in North Italy

**DOI:** 10.3390/ijerph18041812

**Published:** 2021-02-12

**Authors:** Maria Grazia Cagetti, Araxi Balian, Nicole Camoni, Guglielmo Campus

**Affiliations:** 1Department of Biomedical, Surgical and Dental Science, University of Milan, 20142 Milan, Italy; maria.cagetti@unimi.it (M.G.C.); n.camoni@gmail.com (N.C.); 2Department of Restorative, Preventive and Pediatric Dentistry, University of Bern, Freiburgstrasse 7, 3012 Bern, Switzerland; guglielmo.campus@zmk.unibe.ch; 3Department of Surgery, Microsurgery and Medicine Sciences, School of Dentistry, University of Sassari, Viale San Pietro 3/c, 07100 Sassari, Italy; 4School of Dentistry, Sechenov University, 119991 Moscow, Russia

**Keywords:** COVID-19, dentistry, public health services, Italy

## Abstract

A retrospective study was performed to verify if the number of admissions for urgent dental care in the Urgent Dental Care Service of San Paolo Hospital in Milan (Italy) was directly related to the different phases of the COVID-19 pandemic. Different periods were analyzed: 25 March–5 April 2019 (pre-COVID); 23 March–3 April 2020 (lockdown); 8 June–19 June 2020 (reopening); and November 9–November 20 (second wave). Raw data regarding admissions, diagnoses, and treatments were extracted. Descriptive and bivariate analyses were performed. The survey included 901 admissions, 285 in pre-COVID, 93 during lockdown, 353 in reopening, and 170 in the second wave. In each time period, statistically significant differences were found in the prevalence of each kind of diagnoses (χ^2^_(3)_ = 20.33 *p* = 0.01 for endodontic emergencies, χ^2^_(3)_ = 29.05 *p* < 0.01 for cellulitis/phlegmon, χ^2^_(3)_ = 28.55 *p* < 0.01 for periodontal emergencies, Fisher’s Exact Test *p* < 0.01 for trauma, and χ^2^_(3)_ = 59.94 *p <* 0.01 for all other kinds of diagnosis). A remarkable increase in consultations (+186.36%) and other treatments (+90.63%) occurred during reopening. Tooth extraction was the most frequently delivered treatment, but suffered the largest reduction during lockdown (−79.82%). The COVID-19 pandemic has highly affected dental activity in north Italy, underling the weaknesses of a private dental system in a pandemic scenario.

## 1. Introduction

In January 2021, Italy held one of the highest numbers of COVID-19 cases worldwide (2.52 M, total cases) and one of the European Countries with the highest number of deaths with a mortality rate (CFR) of 3.5% [1,2]. Most cases are being diagnosed in the northern regions of the country, Lombardy and Veneto in particular, where the incidence rate of new Sars-Cov-2 infections have raised over 5000 per 100,000 people and the Intensive Care Unit capacity of the National Health Service remains at critical levels [2,3]. During the first wave, the lockdown period including March and April 2020, dental treatments were generally suspended, except for emergency care. No further restrictions have been imposed by the Italian authorities after this first pandemic wave on dental activities.

There has been a significant reduction in the fraction of the gross national product that is spent for the Italian public health system and oral health care is one of the most penalized health sectors. As a consequence, dental care is largely provided by private practitioners and mainly financed by the direct payment of patients or to a lesser extent, by private insurance schemes. In this situation, inequalities in terms of oral health are largely present in the Italian population, producing unmet needs for both children and adults [4]. A recent review paper highlighted persistent inequality in access to healthcare services in different European countries [5]. Unmet healthcare needs, with regard especially to mental and dental health, combined to initiate an overuse of emergency services and an underuse of primary healthcare services [6].

In oral care, making a treatment decision is a demanding task for a dental professional, resulting in considerable variation in treatment practices [7]. Recent EU healthcare policies have encouraged patients to ask questions of their practitioners and to take an active part in decisions regarding their treatment [8,9,10]. The traditional or paternalistic model based on the presumption that the clinician knows best for the patient is argued to have been largely supplanted [11]. Moreover, the profession of the dentist has largely changed, as an increasing number of dentists no longer own dental offices, but are employed by financial structures that own dental clinics. This process has deeply modified the relationship between the patient and the dentist [12]. 

The American Dental Association defines dental emergencies as potentially life threatening that require immediate treatment to stop ongoing tissue bleeding or alleviate severe pain or infection. As part of the emergency guidance, the Association has added urgent dental care, which focuses on the management of conditions that require immediate attention to relieve severe pain and/or risk of infection [13]. A survey performed among Italian dentists during the first wave of COVID-19 showed that the most common urgent procedures provided were the treatment of pulpitis, dental abscess, and prosthesis re-cementation [14]. 

The hypothesis behind this investigation was that the number of admissions for urgent dental cares was directly related to the different moments of the pandemic. To verify this hypothesis, a retrospective study on the admission charts of a public Urgent Dental Care Service in Milan (Lombardy, north Italy) was performed by analyzing the number of admissions, reasons of admissions (oral diagnosis), and delivered treatments and comparing these data to those recorded in a period before the spread of COVID-19.

## 2. Materials and Methods 

This retrospective single-center study was conducted as a clinical audit and, as such, no approval by an Ethics Committee was sought. The study was focused on the activity of the Urgent Dental Care Service (UDCS) of San Paolo Hospital, which serves approximately 500,000 residents in the south area of Milan metropolitan city (about 15% of the population of Milan) and is on average visited by 134 people per week [15,16,17,18]. 

The study was designed as a comparison of the oral health services provided by the UDCS in four different periods of two weeks each: 25 March–5 April 2019 or the pre-COVID period, as the control; 23 March–3 April 2020 or the lockdown period, representing the first wave of SARS-Cov-2 spread in Italy; 8 June–19 June 2020 or the reopening, when most businesses resumed in Italy after the end of the travel ban; and 9 November–20 November or the second wave, when in Lombardy, a new lockdown was declared due to the persistent worsening of COVID-19 infections and death rate.

Data from patients who had visited the UDCS during the four periods were extracted from the Galileo Emergency^®^ (Dedalus Italia S.p.a, Firenze, Italy) medical management software and exported to an Excel spreadsheet (Microsoft Office 365^®^, Redmond, Washington, USA). Data were retrieved anonymously by the Finance and Accounting Division of ASST Santi Paolo e Carlo Hospital and authorized by the Chief Financial Officer for research purposes. Patient information such as age, sex, nationality, and date of visit were collected. Patients who decided to leave the emergency rooms before being seen by a dentist were excluded from further analysis.

Raw data regarding diagnoses and treatments performed on each patient were extracted as the codes registered at the emergency and derived from the International Classification of Diseases, Ninth Revision, Clinical Modification (ICD-9-CM) [19]. The codes were then merged into macro-areas in order to reduce statistical dispersion due to low frequencies. For oral diagnoses the following five macro-areas were identified: endodontic emergences (irreversible pulpitis, acute/chronic periapical periodontitis, dental abscess); cellulitis/phlegmon (extensive dental infections involving oral and perioral soft and bone tissues with face swelling, fever and/or weakness); periodontal emergencies (gingivitis, chronic generalized periodontitis); trauma (dental trauma and/or trauma involving oral mucosa, maxilla and/or mandible); and other (jaw bone disease, oral tumor, trigeminal neuralgia, mucositis/stomatitis due to no-dental infections or systemic disorders such as candidiasis, *herpes simplex* infection, lichen planus, head and neck radio/chemotherapy side effects). Regarding delivered treatments, the following six macro-areas were identified: consultation; tooth extraction (simple and surgical); conservative treatment (restoration, root canal treatment); intraoral radiograph (periapical, bitewing); panoramic radiograph; and other (dental prothesis and/or orthodontic appliances repair, scaling, root planning, biopsy, cone-beam computer tomography, drug administration, blood tests).

The number of SARS-Cov-2 new cases in Lombardy was collected for the above-mentioned periods based on the COVID-19 report released every day at 6 PM (UTC +1 h) by the Italian Civil Protection Department [20].

Admissions at the UDCS were divided in three age groups: children (≤18 years), adults (≥19 ≤ 65 years), and elderly (>65 years). Descriptive analysis was performed using the same software, while bivariate analysis was performed using the free-ware statistical web-software Vassar Statistics [21]. Chi-square was performed for multiple comparison of patients’ age and sex. Chi-square proportion was performed for multiple comparison of oral diagnosis and treatments. Fisher’s exact test and Bonferroni correction were performed when a cell had a value less than 5. One-way-Analysis of Variance (ANOVA) was used to compare the mean number of admissions and mean number of new Covid-19 cases recorded in the different periods.

## 3. Results

The survey included data of 901 admissions managed at the UDCS of San Paolo Hospital in Milan, 285 in the pre-COVID period, 93 during lockdown, 353 in the reopening, and 170 in the second wave. A decrease in admissions was observed during lockdown and the second wave periods of −67.37% and −40.35%, respectively, when compared to the pre-COVID period. The number of admissions was statistically significantly associated (*p* < 0.01) with the mean number of new cases of COVID-19 recorded in each period (*data not in tables*). During the reopening, the number of patients who requested urgent dental care increased + 23.86% compared to the control period (pre-COVID). The mean age of the patients was 43.74 ± 21.70 years in the pre-COVID, 46.65 ± 17.74 in the lockdown, 39.90 ± 23.63 in the reopening, and 40.65 ± 22.38 years in the second wave. The Italian nationality represented the majority of patients in all the considered periods with a statistically (χ^2^_(4)_ = 18.43 *p <* 0.01) significant increment during the lockdown period *(data not in table**).*


Overall, the number of admissions at the UDCS showed a specific trend that was inversely associated with the SARS-CoV-2 pandemic evolution (Pearson’s *r* two-tailed test, *p* < 0.01) in all age groups, as displayed in Figure 1. This trend was also statistically significant for both females (χ^2^_(6)_ = 22.70 *p* < 0.01) and males (χ^2^_(6)_ = 14.78 *p* = 0.02) (*data not in table*). Overall, a higher number of males attended the UCDS in all age groups and time periods. The relative percentage of female and male patients did not change significantly (*p* > 0.05) within the child and adult groups in the different time periods (Table 1); on the other hand, male to female ratio reversed in the elderly group in the reopening and second wave, even if it did not reach the statistical significance (*p* = 0.09). The relative percentage of children, adult, and elderly who accessed the UDCS did not change significantly (χ^2^_(6)_ = 5.65 *p* = 0.13) during the considered periods. A remarkable peak in children’s admissions was observed in the reopening (+140.54%).

Oral diagnosis performed during the four periods is shown in Table 2. In each time period, statistically significant differences were found in the prevalence of each kind of diagnoses (χ^2^_(3)_ = 20.33 *p* = 0.01 for endodontic emergencies, χ^2^_(3)_ = 29.05 *p* < 0.01 for cellulitis/phlegmon, χ^2^_(3)_ = 28.55 *p* < 0.01 for periodontal emergencies, Fisher’s exact test *p* < 0.01 for trauma and χ^2^_(3)_ = 59.94 *p <* 0.01 for all other kinds of diagnosis). A reduction in the number of oral diagnoses was observed both in the lockdown and second wave periods, which was less marked in the latter. Endodontic emergencies were the most performed diagnosis in each considered period and accounted for the highest decrease (−70.62%) during the lockdown (Table 1). Traumas significantly decreased during lockdown (−83.33%) and the second wave (−75.00%), while an increase was observed in cellulitis/phlegmon (+73.33%), periodontal emergencies (+25.64%), and other oral diagnosis (+69.39%) during the reopening period (Table 1).

All treatments/diagnostic exams were reduced significantly both in the lockdown (χ^2^_(1)_ = 11.73 *p* < 0.01) and in the second wave (χ^2^_(1)_ = 6.48 *p* < 0.05) compared to pre-COVID, while a not significant increase was observed in the reopening (χ^2^
_(1)_ = 1.20 *p* > 0.05). Statistically significant differences in the prevalence of each kind of treatments/diagnostic exams in each time periods were found (χ^2^_(3)_ = 26.57 *p* < 0.01 for consultations, χ^2^_(3)_ = 26.57 *p* < 0.01 for tooth extraction, χ^2^_(3)_ = 35.3 *p* < 0.01 for restorative/endodontic treatments, χ^2^_(3)_ = 25.08 *p* < 0.01 for intraoral radiographs, χ^2^_(3)_ = 14.56 *p* < 0.01 for panoramic radiograph, and χ^2^_(3)_ = 37.95 *p* < 0.01 for all other kind of treatments). A reduction of all treatments and/or diagnostic exams was observed both during the lockdown and second wave, which was again, less marked in the latter. Tooth extraction was the most frequently delivered treatment in each period, but also suffered the largest reduction (−79.82%) from pre-COVID to lockdown (Figure 2). A remarkable increase in consultation (+186.36%) and other treatments (+90.63%) occurred during the reopening. Intraoral radiographs showed a strong reduction in the lockdown (−74.42%) and did not reach the pre-COVID values even during the reopening period (−44.96%). In contrast, panoramic radiographs, after 50% reduction during lockdown, increased in the reopening (+37.5%) and finally suffered a slight reduction in the second wave (−10.41%).

Overall, diagnostic exams (intraoral and panoramic X-ray) together with tooth extractions were consistently the most delivered procedures in all time periods (Figure 2).

## 4. Discussion

The present investigation showed that significant modifications in the number of admissions, oral diagnosis, and treatments performed occurred in the UDCS of San Paolo Hospital in Milan (Italy) during the COVID-19 pandemic. These changes were significantly associated with the SARS-Cov-2 epidemiological trend and restrictions imposed by the Italian authorities.

In the present study, the number of patients seeking urgent dental care was inversely associated with the Sars-Cov-2 epidemiologic spread and this trend was observed both in females and males regardless of age. As a consequence, a significant drop in admissions occurred during the lockdown and second wave periods, while a rebound was registered in the reopening period. To the best of the authors’ knowledge, no previous studies have investigated the trend of urgent dental care provided during the different pandemic phases. Emergency admissions were strongly reduced during the COVID-19 outbreak worldwide, as suggested by previous studies reporting a variable reduction in visitors ranging from 38% to 80% [22,23]. National and international guidelines released by the different Dental Associations have advocated limiting dental activities to emergencies and urgent life-threatening conditions in order to encourage social distancing [13,24,25]. It was reported that private dental practitioners had coped with many dental urgencies through phone triage, sending drug prescriptions via email, and conducting online consultation via smartphones, tablets, or PCs [26]. The UDCS of San Paolo Hospital did not switch to a remote model workflow for dental emergency care during the COVID-19 outbreak. The reduction in patients recorded during lockdown and the second wave in the UDCS could be a consequence of either a raising awareness about what should be considered a dental emergency and a widespread fear of going to an emergency department and/or hospital [27]. Dental activities have since struggled to recover because of ongoing economical and healthcare difficulties such as shortages of personal protective equipment and delayed admissions in dental offices due to the COVID-19 restrictions [14,28]. The peak registered during the reopening might be the consequence of a partial incapacity of the private dental healthcare system to meet patient needs during the first pandemic recovery.

Generally, more male patients attended the UDCS in the pre-pandemic period and this trend was confirmed during the COVID-19 outbreak. Nonetheless, an inverse tendency was noticed in the elderly group during the reopening and second wave compared to pre-COVID and lockdown. Male and old people have been proven to face severe respiratory diseases more frequently due to Sars-Cov-2 infection, as revealed by the higher number of intensive care unit (ICU) admissions and mortality rate [29,30]. The raising awareness and acknowledge about COVID-19 clinical features, regardless of the actual spread of coronavirus infection, might have induced old male people to follow social distancing rules more, thus also avoiding visiting the UDCS if not strictly necessary during the reopening and second wave. 

The peak of children’s admissions recorded during the reopening compared to the control period (pre-COVID) occurred at the end of school activities, but this cannot fully explain such an increase. Outpatient dental care was strongly reduced during lockdown and struggled to recover the following months as there were particular concerns about aerosol generating procedures (AGP) such as teeth cleaning, sealants, and restorative treatments. Italian and international guidelines have recommended avoiding AGP as much as possible, and to choose alternative treatments or postpone therapies whenever possible [13,24,25,28,31]. This might have influenced children’s oral health needs, since they no longer had easy access to free dental treatments provided by the Regional Health System (RHS) in Lombardy (i.e., preventive, restorative therapy) and found difficulties in turning toward private practitioners. Adults and the elderly are supposed to have been less affected from this point of view, as they are usually forced to go to private practitioners since they are not covered by the RHS for dental care with the exception of a few categories of patients with serious illness and/or socio-economic issues. Minimally invasive methods and preventive measures suggested to reduce AGP in dental practice during the coronavirus pandemic could be of valuable interest, in order to both meet the needs of patients and provide adequate answers to those who cannot afford the costs of private dental [32].

Oral diagnoses and dental treatments significantly reduced during lockdown and reopening, which was mainly due to the drop in endodontic emergencies and tooth extractions, which represent the most prevalent categories in all time periods, as previously reported in a comparable dental setting in both the pre-pandemic and during COVID-19 outbreak [22,26,33,34,35,36]. More than 80% of dental care in Italy is provided by private practitioners; moreover, less than 50% of Italian citizens aged 15–65 years undergo at least one dental check-up per year according to the 2017 Italian National Institute of Statistic Report [37]. The more people age, the less access to a dental healthcare system they have: less than a third of people aged over 75 years regularly go to the dentist and the lowest-income people aged over 65 are more likely to undergo tooth extraction as treatment [37]. The results of the present study suggest that a high number of endodontic emergencies are actually the first dental consultation related to dental caries, while tooth extraction often represents the extreme answer for those who refused and/or cannot afford restorative and rehabilitative treatments. Previous literature has reported on how socio-economic status affects patient decision-making in oral and medical health [38,39,40], and the COVID-19 pandemic has further widened social disparities [41].

The prevalence of oral/facial trauma also showed a wide reduction during lockdown and reopening. Dental and oral tissues injuries often occur at school, workplaces, and/or during play activities or sports [42], so it is not surprising that we observed a reduction when strict restrictions to people’s movements were applied.

Cellulitis/phlegmon showed the least reduction during lockdown as they are associated with worrying symptoms such as swelling, fever, weakness, severe toothache, making patients more likely to seek urgent dental care even in a pandemic scenario [22,43].

Many preventive dental treatments such as regular tooth-cleaning and periodical check-ups were cancelled during the two-month long Italian lockdown, and they have been gradually restored in the ongoing months due to the high risk related to aerosol generating procedures (AGP) and/or the supposed lack of urgent need. In the same period, an increased number of consultations and other types of treatments (i.e., orthodontic appliances repair, biopsy) has been reported. These findings highlight the essential role of preventive dental care and implies a deep afterthought for future perspectives. 

Intraoral radiographs were strongly reduced during lockdown and progressively increased in the following periods, even if they did not reach pre-COVID values, while panoramic radiographs slightly increased during the coronavirus pandemic in all time periods. Dentists preferred to perform extra-oral radiographic techniques that did not require oral tissue manipulation, thus reducing the risk of cross-infection, as recommended by scientific and dental associations [13].

This retrospective survey has some limitations. First, the patients’ clinical history was unfeasible since their medical charts were not available in the management software used at the UDCS. Moreover, it was not possible to differentiate the first admission from follow-ups and/or further access of the same patient; this might have caused a bias in the reported results.

## 5. Conclusions

The results of this retrospective investigation on admissions and dental care at an UDCS in Milan suggests that the COVID-19 pandemic has highly affected dental activities in Italy. 

The number of admissions was inversely associated with the COVID-19 pandemic evolution: significantly less patients sought urgent dental care during lockdown and the second wave. A rebound has instead been observed during the reopening.

A significant reduction in endodontic emergencies and tooth extractions was reported during lockdown and the second wave, suggesting that many patients usually sought a first dental consultation and an affordable treatment, rather than primary urgent care in the UDCS.

A significant increase in consultations and other types of treatments (orthodontic appliances repair, biopsy) was observed during reopening, unveiling the weaknesses of the predominantly private dental system in a pandemic scenario.

The findings of the present study, with all the inner limitations, reiterate the essential role of preventive actions in dental care and should be considered for future policies in this field.

## Figures and Tables

**Figure 1 ijerph-18-01812-f001:**
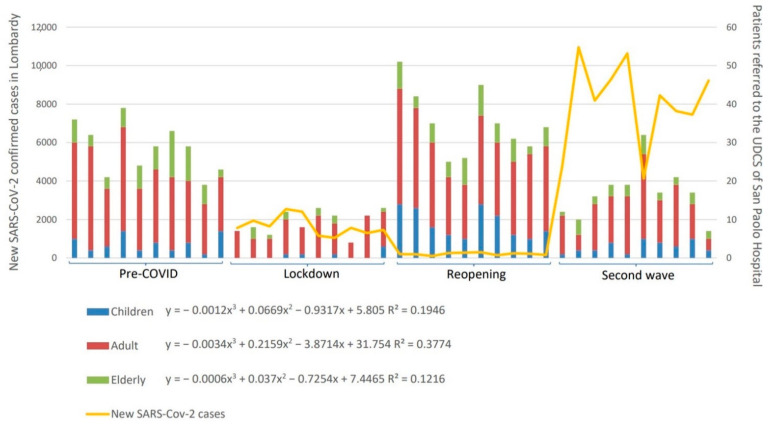
Curve of the number of patients visiting the Urgent Dental Care Service (UDCS) in the four time periods and new cases of SARS-CoV-2. Trend coefficient are reported for children (≤18 years), adults (19–65 years), and the elderly (>65 years).

**Figure 2 ijerph-18-01812-f002:**
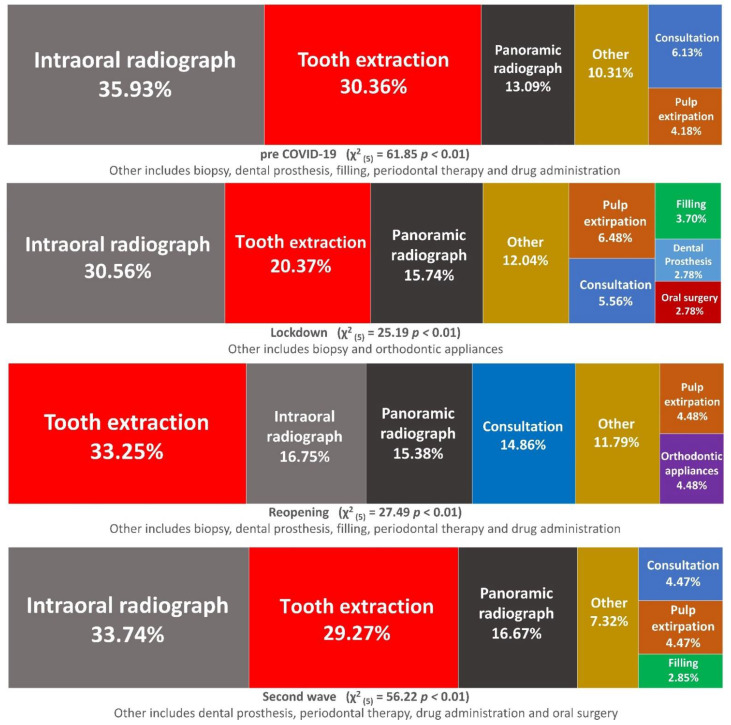
Prevalence of treatments and diagnostic exams performed in the different considered periods. Treatments that accounted for less than 2.5% each were grouped into the “Other” category.

**Table 1 ijerph-18-01812-t001:** Age and sex of patients attending the Urgent Dental Care Service (UDCS) of San Paolo Hospital in Milan during the four time periods considered.

Time Periods	≤18 Years		19–65 Years		>65 Years		
	Female n (%)	Male n (%)	Female n (%)	Male n (%)	Female n (%)	Male n (%)	TOTAL
							n (%)
pre-COVID	10 (16.13%)	27 (27.27%)	78 (30.59%)	113 (33.43%)	23 (31.94%)	34 (45.33%)	285 (28.63%)
Lockdown	1 (1.61%)	5 (5.05%)	32 (12.55%)	44 (13.02%)	3 (4.17%)	8 (10.67%)	93 (10.32%)
Reopening	41 (66.13%)	48 (48.48%)	96 (37.65%)	116 (34.32%)	31 (43.06%)	21 (28.00%)	353 (39.18%)
Second wave	10 (16.13%)	19 (19.19%)	49 (19.22%)	65 (19.23%)	15 (20.83%)	12 (16.00%)	170 (18.87%)
TOTAL	62 (100.00%)	99 (100.00%)	255 (100.00%)	338 (100.00%)	72 (100.00%)	75 (100.00%)	901 (100.00%)
	Fisher’s ET *p* = 0.13		χ^2^_(3)_ = 0.84, *p* = 0.84		Fisher’s ET *p* = 0.09		

**Table 2 ijerph-18-01812-t002:** Oral diagnoses performed during the four time periods and column and row statistical analysis. Percentages are calculated in columns, except for the total column in rows.

Oral Diagnosis							
Time Periods	Endodontic Emergencies	Cellulitis/Phlegmon	Periodontal Emergencies	Trauma	Other	TOTAL	Statistical Analysis
	n (%)	n (%)	n (%)	n (%)	n (%)	n (%)	
pre-COVID	143 (32.35%)	30 (25.21%)	39 (31.97%)	24 (47.06%)	49 (30.26%)	285 (28.63%)	χ^2^_(4)_ = 73.90 *p* < 0.01
Lockdown	42 (9.50%)	22 (18.49%)	14 (11.48%)	4 (7.84%)	11 (7.24%)	93 (10.32%)	χ^2^_(4)_ = 61.51 *p* < 0.01
Reopening	151 (34.16%)	52 (43.70%)	49 (40.16%)	17 (33.33%)	83 (51.32%)	353 (39.18%)	χ^2^_(4)_ = 51.98 *p* < 0.01
Second wave	106 (23.98%)	15 (12.61%)	20 (16.39%)	6 (11.76%)	23 (11.18%)	170 (18.87%)	χ^2^_(4)_ = 143.73 *p* < 0.01
TOTAL	442 (100%)	119 (100%)	122 (100%)	51 (100%)	167 (100%)	901 (100.00%)	
Statistical analysis	χ^2^_(3)_ = 20.33 *p* = 0.01	χ^2^_(3)_ = 29.05 *p* < 0.01	χ^2^_(3)_ = 28.55 *p* < 0.01	Fisher’s ET *p* < 0.01	χ^2^_(3)_ = 59.94 *p* < 0.01	χ^2^_(3)_ = 26.57 *p* < 0.01	

## Data Availability

The raw data are available on request. Please contact the corresponding author.

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
