# Peer review of "Influence of the COVID-19 Pandemic on Dental Emergency Admissions in an Urgent Dental Care Service in North Italy"

_ijerph, 2021, doi:10.3390/ijerph18041812_

Round 1

Reviewer 1 Report

Interesting paper:

Abstract: Please increase the results section. It is excessive short.

Introduction:

Unmet healthcare needs regard especially mental and dental health, combined to an overuse of emergency services and an un-deruse of primary healthcare services. Please consider this reference: PMID: 33419107.

Moreover, in the European Mediterranean countries, the 55 profession of the dentist has largely changed, as an increasing number of dentists no 56 longer own dental offices, but are employed by financial structures that own dental clin-57 ics. This process has deeply modified the relationship between the patient and the den-58 tist.  Please add reference.

Results: Well presented. Only improve the quality of the pictures, especially Figure 2.

Discussion:

Limitations?

Please check reference´s style.

Author Response

Dear Reviewer 1,

Thank you for helping us in improving the manuscript.

Please find below our answers to your comments/suggestions.

Reviewer 2 Report

I read with great interest the manuscript entitled “Influence of the COVID-19 pandemic on dental emergency admissions in an Urgent-Dental-Care-Service in North-Italy”.

The authors of the manuscript must make some modifications in order to consider their article for publication in this journal:

1) In the abstract of the manuscript, the centre where the project was carried out must be indicated.

2) Was approval by a Bioethics Committee sought for the study?

3) Lines 47-49. The bibliographical reference is duplicated.

4) The main limitations of the study should be included in the discussion.

5) The conclusions should be modified to include some data on the results of the study.

6) The authors should revise the bibliography. Some bibliographical references are not described according to the journal's regulations.

Author Response

Dear Reviewer 2,

Thank you for helping us in improving the manuscript.

Please find below our answers to your comments/suggestions.

Reviewer 3 Report

Dear Author:

It's a very interesting topic. Overall the present manuscript is very well written. There are no revisions in the Abstract, Introduction, Result, and Discussion sections. In the Materials and Methods section, the approval and number of the Institutional Review Board must be inserted.

Best regards,

Author Response

Dear Reviewer 3,

Thank you for helping us in improving the manuscript.

Please find below our answers to your comments/suggestions.

Round 2

Reviewer 2 Report

The authors should modify the resolution of figure 1 for a better reading of the manuscript.